# Androgen Receptor Expression in Thai Breast Cancer Patients

**DOI:** 10.3390/medsci4030015

**Published:** 2016-09-14

**Authors:** Suthat Chottanapund, M. B. M. Van Duursen, Kumpol Ratchaworapong, Panida Navasumrit, Mathuros Ruchirawat, Martin Van den Berg

**Affiliations:** 1Division of Environmental Toxicology, Chulabhorn Graduate Institute, Bangkok 10210, Thailand; panida@cri.or.th (P.N.); mathuros@cri.or.th (M.R.); 2Laboratory of Environmental Toxicology, Chulabhorn Research Institute, Bangkok 10210, Thailand; 3Center of Excellence on Environmental Health, Toxicology and Management of Chemicals, Bangkok 10400, Thailand; 4Bamrasnaradura Infectious Diseases Institute, Ministry of Public Health, Nontaburi 11000, Thailand; 5Institute for Risk Assessment Sciences, Utrecht University, Utrecht NL-3508, The Netherlands; m.vanduursen@uu.nl (M.B.M.V.D.); m.vandenberg@uu.nl (M.V.d.B.); 6Department of Surgery, Charoenkrung Pracharuk Hospital, Bangkok 10120, Thailand; siajeab@hotmail.com

**Keywords:** breast cancer, androgens, androgen receptors

## Abstract

The aim of this study was to investigate prevalence and related factors of androgen receptor (AR) expression in Thai breast cancer patients. A descriptive study was done in 95 patients, who were admitted to Charoenkrung Pracharak Hospital, Bangkok (2011–2013). Statistical relationships were examined between AR protein expression, tumor status, and patient characteristics. Compared with those from Western countries, ethnic Thai patients were younger at age of diagnosis and had a higher proliferative index (high Ki-67 expression), which indicates unfavorable prognosis. In addition, 91% of the Thai breast tumors that were positive for any of the following receptors, estrogen receptor (ER), progesterone receptor (PR), and human epidermal growth factor receptor 2 (HER2) also expressed the AR protein, while in triple negative breast tumors only 33% were AR positive. ER and PR expression was positively related with AR expression, while AR expression was inversely correlated to Ki-67 expression. AR status was strongly correlated with ER and PR status in Thai patients. There is an inverse relationship between Ki-67 and AR, which suggests that AR may be a prognostic factor for breast cancer.

## 1. Introduction

Breast cancer is by far the most common type of cancer among women around the world. In 2012, there were about 1,671,000 new cases of female breast cancer worldwide [1]. According to the American Cancer Society, there still is an increasing trend of female breast cancer incidence. The number of female breast cancer cases in the US in 2015 was 231,840, which accounts for 122.8 per 100,000 persons/year of all female cancers [2]. The number of new breast cancer cases in Thailand in 2012 was about 12,000 cases, which accounts for 137.5 per 100,000 persons/year [3]. Interestingly, the mortality of breast cancer patients is strongly dependent on region and ranges from 6 to 20 per 100,000 persons/year [1].

There is scientific consensus that breast cancer is a systemic disease and multimodality treatment is needed to cure or prevent residual cancer after surgery [4,5]. The three well-established systemic treatments for breast cancer are chemotherapy, hormonal therapy and biological targeted therapy. Systemic chemotherapy is generally considered to have the highest efficacy against breast cancer, but it can cause serious side effects for the patient [6,7]. Compared to chemotherapy, hormonal treatment has fewer side effects, but it can only be used if the breast tumor cells express the estrogen receptor (ER) [4,8]. Currently, there are two types of hormonal treatments available for ER responsive breast cancer, which use either an anti-ER drug (e.g., tamoxifen) [9] or an aromatase inhibitor (e.g., letrozole) [10]. In addition, biological targeted therapies are a relatively new way to fight breast cancer and they target a specific protein in the tumor cell. However, the costs of these treatments are very high and often tumor specific, which makes them less affordable for developing countries and emerging economies. Despite the availability of these different systemic treatment options, breast cancer continues to have a high case-fatality of about one-fifth of all cases [2].

Mortality and morbidity are higher for patients with triple negative breast cancer, which has a negative ontology for hormonal receptors [11,12]. At present, chemotherapy is the only systemic treatment for triple negative breast cancer, but the results on overall survival still remain poor [13,14]. Most of the patients with a triple negative breast tumor will develop brain metastasis in the terminal phase of the disease [14]. Therefore, there is an urgent need to find systemic treatment therapies that can increase overall survival for these patients.

There is growing evidence that the androgen receptor (AR) might be a new target for systemic breast cancer treatment [15,16]. Several studies show that androgens can inhibit mammary epithelial proliferation and breast tumor growth. Conversely, some androgens may actually increase tumor size due to their conversion into estrogens by aromatase and their subsequent ability to bind to the ER, which leads to increased proliferation of the ER-responsive breast tumor cells [16]. We previously showed, using a co-culture model of human breast tumor cells with primary breast fibroblasts, that some androgens can indeed inhibit breast cancer cell growth via the AR [17]. However, the positive or negative effects depend on the type of androgen used, e.g., aromatizable testosterone (T) or non-aromatizable dihydrotestosterone (DHT), but also on the ERα and AR status of the breast tumor cell and aromatase activity in the surrounding breast fibroblasts [17]. Our present study was aimed to determine the prevalence of AR expression and its relationship with some clinical and pathological parameters in Thai breast cancer patients.

## 2. Methods

### 2.1. Thai Breast Cancer Patient and Tumor Characteristics

This exploratory study enrolled a total of 95 Thai breast cancer patients that were admitted in the Charoenkrung Pracharuk Hospital (Bangkok, Thailand) between October 2011 and October 2013. Clinical and pathological data were collected from the patients’ records. These data included age, menstrual status, body mass index (BMI), pre-operative pathology, previous diagnostic method, tumor size, tumor location, clinical nodal status, type of surgery, tumor stage, final pathology, pathological nodal metastasis, nuclear grade, margin of lymphovascular invasion (LVI), neural invasion, proliferative index (Ki-67), status of the ER, progesterone receptor (PR) and human epidermal growth factor receptor 2 (HER2) responsiveness. The immunohistochemistry staining process was done with a Bench Mark GX automatic machine (Ventana, Roche, Tucson, AZ, USA). The level of receptor expression was reported by a Thai board qualified pathologist. Rabbit monoclonal primary antibodies (clone SP1), rabbit monoclonal primary antibodies (clone 1E2), anti-HER2 (4B5) rabbit monoclonal primary antibodies, and rabbit monoclonal primary antibodies (clone 30-9) (Ventana, Roche) were used to stain ER, PR, HER-2 and Ki-67, respectively.

### 2.2. Ethical Considerations

All patients who enrolled in the study gave a written informed consent. In addition, these patients received full information and explanation about the study objectives and rights that the patient has before enrolling in the study. The study was approved by the Ethics Committee for Researches Involving Human Subjects at the Bangkok Metropolitan Administration (Charoenkrung Pracharuk Hospital), on 21 October 2011, with the project identification number 073.54. This research has been carried out in accordance with the Code of Ethics of the World Medical Association (Declaration of Helsinki).

### 2.3. Androgen Receptor Protein Expression

The presence of the androgen receptor protein was tested in the remaining pathological specimens of the breast tumor tissue by using an anti-androgen receptor (SP107) rabbit monoclonal primary antibody (Cell-Marque, Roche, CA, USA), which was stained with the immunoperoxidase method [18]. The immunohistochemistry staining process was done with a Bench Mark GX automatic machine (Ventana, Roche). The level of androgen receptor expression was reported by a Thai board qualified pathologist. While immunohistochemistry is a common method for protein detection, there is no standard cutoff point for androgen expression in breast cancer [19,20]. Therefore, we adopted the AR staining procedure that is used for prostate cancer and the H-score method [21], which is similar to the immunohistochemical scoring methods that are used for other hormone receptors [21,22,23]. This score was given as multiplication of the intensity level of staining (0 = none, 1 = weak, 2 = moderate, and 3 = strong) by the percent of staining (the total score is 3 × 100% = 300) [21,22,23]. Figure 1a,b show typical examples of AR^+^ and AR^−^ breast tumor tissue from Thai patients in this study.

### 2.4. Sample Size Calculation for Detecting the Prevalence of AR Expression in Thai Patients

To achieve our primary research objective, the following sample size formula for single proportion was applied:
$$n=\;\frac{z^{2}\;\left(p\right)\left(1\;-\;p\right)\;}{d^{2}}$$
*z* = 1.96 at alpha 0.05% or 95% confidence interval*p* = Proportion of androgen receptor positive; 70%, according to Ren et al. [24]*d* = Acceptable error of 10% = 0.01*n* = (1.96 )^2^ · (0.7) · (1−0.7)/(0.1) · 2 = 81 cases

This calculation indicated that at least 81 patients had to enroll in this study in order to achieve enough statistical power.

### 2.5. Data Analysis

The statistical analysis of the clinical and pathological data was done with descriptive and inferential statistics. For descriptive analysis, the total values and percentages were used as categorical variables and the mean value with standard error of the mean for the numerical values. In addition, a comparative analysis was carried out using bivariate and multivariate analysis; the χ^2^ test was used for categorical variables and Student’s *t* test was used for numerical variables with normal distribution. A *p* value <0.05 was considered statistically significant. Logistic binary regression was used for the multivariate analysis.

## 3. Results

### 3.1. Patient and Tumor Characteristics and AR Expression

A total of 95 ethnic Thai breast cancer patients were enrolled in this study. Their mean age was 51.37 ± 1.17 (mean ± standard error). Mean body surface area (BSA) and BMI were 1.56 ± 0.01 and 24.38 ± 0.46, respectively. Forty-one cases (43.16%) were in natural post-menopausal state. Axillary nodes were clinically palpable in twelve cases (12.63%), while 46 cases (48%) were pathologically node positive.

Invasive ductal carcinoma was the most common diagnosis in these patients (95.79%) and about 56% of the cases had lympho-vascular invasion. About 70% of the patients had a high proliferative index (Ki-67 expressed ≥20%). The pathological staging at the time of treatment was equally distributed with about 30% of the patients in a late stage of disease (Stage 3).

Approximately 60% of the malignant breast tumors were ER and PR positive, 40% had HER-2 receptor expression (HER-2 positive 2+ and 3+), while only 19% were of triple negative ontology. Of all tumors, 80% (76/95) expressed the AR protein. For more detailed information, see Figure 2 and Figure 3.

### 3.2. Association of Tumor Characteristics with AR Expression

91% of the breast tumors with positive expression of ER, PR, or HER2 also expressed the AR. In contrast, only one-third (6/18) of the triple negative tumors showed an expression of the AR. The factors (including baseline characteristics, pathological and hormonal factors) related to AR expression were statistically analyzed. The expression of the AR was strongly related to ER expression. In addition, it was found that a triple negative tumor status was inversely related to the expression of the AR (*p* value <0.001) (Table 1).

### 3.3. Relationship between AR Expression (H-Score) and Other Hormonal Receptors and Ki-67

The H-score for AR expression was positively correlated with expression of the ER, PR and HER-2 (*p* < 0.001, 0.014, and 0.006, respectively). However, in the triple negative tumors, the H-score of the AR was approximately threefold lower than in the hormonal positive tumors (*p* < 0.001). These quantitative differences in the H-score of AR expression with expression of other (hormonal) receptors are shown in Figure 4a–d. The relationship between AR^+^ and AR^−^ tumors with the cell proliferation marker Ki-67 was also analyzed, and it was found that AR^−^ tumors had an approximate 50% higher Ki-67 expression than AR^+^ tumors (*p* < 0.01) (Figure 5).

## 4. Discussion

This study is the first to describe AR protein expression in Thai breast cancer patients. Demographic data of Thai breast cancer patients are clearly different from those of Western patients. Unlike those from Western countries, breast cancer patients in Thailand generally come to the hospital with a more advanced, late stage of the disease. This is illustrated by our data in which our patients had a higher breast cancer stage than that generally observed in studies from Western countries. In fact, 15 cases in our study needed chemotherapy before surgery (neoadjuvant chemotherapy; all pathological data of this group of patients were obtained from core needle biopsy), representing 16% of the patients. The incidence of neoadjuvant chemotherapy in Western countries is usually less than 10% [3,25]. In our study, Thai breast cancer patients were younger at the age of diagnosis (90% were diagnosed at an age below 65, compared to only 50% of UK cases diagnosed at an age below 65. In addition, they are more often in premenopausal state [2]. The mean age of our patients was 52 years old; half of them were still in the pre-menopausal phase. The most common histological type of breast cancer in our study group was invasive ductal carcinoma and most of our patients were undergoing simple mastectomy (81%). Two-thirds of the tumors expressed hormonal receptors (estrogen and/or progesterone), which is a similar number to that reported from Western countries [26]. In our study, 40% of the tumors also expressed the HER2 receptor, which is a much higher proportion than that reported from Western countries (19%) [27,28]. Approximately 19% of our patients had a triple negative breast tumor, which is comparable to the occurrence reported in Western countries (15%) [29,30]. Most of the Thai patients in this study had a high expression of the proliferative index Ki-67 (68%), which indicates that these patients have a less favorable prognostic histology [31]. These data clearly demonstrate the geographic variation in breast cancer characteristics, which may strongly influence therapeutic success. Clearly, more research is needed to provide better breast cancer treatment options in non-Western countries. This is also reflected in the relatively high mortality rate for breast cancer in Asian countries as compared to Western countries [1]. Here, genetic variations might also play a role. It should be noted that Thailand’s population is relatively heterogeneous culturally, but genetically consists of only a few ethnic groups of which some have Chinese, Vietnamese or Malaysian backgrounds. In this study, we did not establish genetic descent, but considered all women of Thai ethnicity as “Thai”.

There is emerging evidence that the androgen receptor might be an additional target for systemic breast cancer treatment [15,16]; however, the role of androgens and the AR in breast cancer is still controversial. Androgens can either induce or inhibit proliferation of breast cancer. Some clinical studies and in vitro experiments support the view that a low dose of androgens can increase the risk of breast tumor formation and growth [16,32]. In contrast, there are studies that indicate androgens can inhibit the growth of hormonal negative breast cancer cells both in vitro and in vivo, if a strong expression of the AR is present [33,34]. The underlying mechanism for this phenomenon may partly be explained by the conversion of some androgens to estrogens by the aromatase enzyme, which can subsequently stimulate breast tumor cell proliferation [17]. In a clinical study done by Key et al. and our previous in vitro studies with a co-culture of human breast fibroblasts and tumor cells, indications were found that aromatase can play a key role in the indirect stimulation of androgens on cell proliferation [17,35]. However, the latter study also indicated that non-aromatizable androgens can inhibit the growth of breast cancer cells that have AR expression. These androgens can also induce apoptosis regardless of the cells’ ER/PR status. Furthermore, androgens may exert their effects via two other pathways; direct stimulation by binding to AR (AR-positive/ER-negative tumors) or increased synthesis of growth factors, such as epidermal growth factor (EGF) (ER-negative/AR-negative tumors). A more recent study demonstrated that the AR/ER ratio influences breast tumor responses to hormonal treatment [36]. Moreover, it was concluded that the AR antagonist enzalutamide could be used for treatment of AR^+^ tumors regardless of ER status, since it can block both AR- and ER-mediated tumor growth. Clearly, AR and ER status of a breast tumor will determine the effect of an (anti)androgen on breast tumor cell proliferation.

The AR expression in our Thai patients was about 80%, which is comparable to previous reports from Western countries [24,37]. Noticeably, the Thai triple negative breast cancer patients had a lower expression of the AR (33%) compared to Western patients (40%–50%) [15], but AR expression observed in our study was comparable with that reported for Chinese breast cancer patients (28%) [38]. These differential findings between breast cancer types in Asian and Western countries may result from population stratification bias and the difference of demographic data [39,40]. Some studies have reported that the AR expression is also related to the size of the tumor, the spread to lymph nodes in the armpit, certain kinds of histology, the stage of breast cancer and the status of the ER and PR [41,42]. However, these relationships are equivocal, as another study did not detect an association between the AR and any clinical pathological characteristics [43]. In our study, the expression of the AR was significantly associated with the concurrent expression of the ER and PR. Our observation is in agreement with that of Soreide et al. who also reported a high AR expression in hormonal positive breast tumors [44]. Another Asian study done by Ogawa et al. reported that the menstrual status was related to the AR expression [42]. However, we did not observe this in our study population (data not shown).

Another interesting finding in our study was the fact that the expression of the AR was inversely correlated with the non-ER-dependent proliferation index Ki-67. This is an essential protein for progression through the cell cycle [45] and is considered to be a hormone-independent prognostic factor for disease-free and overall survival in breast cancer patients [46]. Based on our observations, there is a possibility that AR expression in hormone-dependent breast tumors may also be a prognostic factor for a favorable outcome of the disease. However, considering the limited time that has passed since the collection of the tissue samples in this study, no conclusions can yet be drawn on clinical outcomes and prognosis in our study population based on AR status. Still, a study by Gasparini et al. showed that AR^+^ breast cancer patients had a better survival prognosis than those who were AR^−^ [19]. In line with these observations, it should be noted that, in the past, androgens have been used to treat advanced stage breast cancer as reported by de Matteis et al. and Scholl et al. [47,48]. In addition, such a treatment was also effectively used to treat skin metastasis of male breast cancer patients by Battelli et al. [49]. However, treatment with androgens was abolished in the past because of serious side effects such as hirsutism and aggressive behavior [50]. Moreover, the success of anti-estrogens [51] and aromatase inhibitors to treat hormonal positive breast cancer patients [52,53] made the use androgen therapy less needed. Nevertheless, more recent results, including those from our study, again indicate a possible role of the AR in inhibiting breast tumor progression. However, more experimental and human studies are needed to define more clearly the relevant target group of patients and tumor characteristics needed for a positive response.

## 5. Conclusions

In our study, the majority of Thai breast cancer patients had tumors that expressed the AR. These tumors were mostly hormonal and HER-2 positive in nature. Expression of the AR in triple negative breast tumors was less common. When compared with data from Western studies, the Thai patients show distinct differences in AR expression and the ratio between hormonal/HER-2 responsive and triple negative breast tumors. An inverse relationship between the non-hormone-dependent proliferation index Ki-67 and AR expression was found in this study. It is suggested that AR expression may be a prognostic factor for a more favorable outcome of the disease. Further experimental and human studies are needed to establish whether or not the AR is also a useful therapeutic target in a specific group of patients besides already available adjuvant systemic treatments with ER antagonists and aromatase inhibitors.

## Figures and Tables

**Figure 1 medsci-04-00015-f001:**
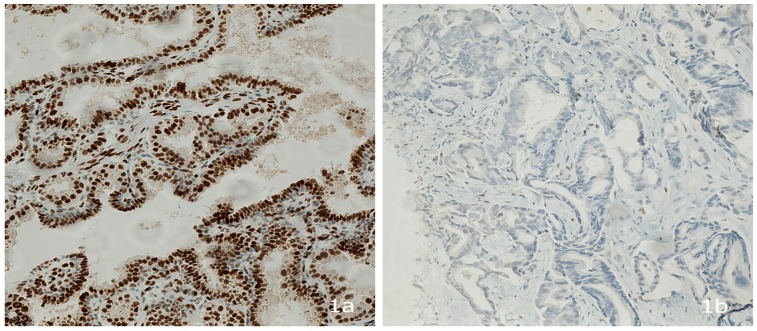
(**a**) Androgen receptor positive breast cancer. Stained androgen receptor in breast cancer tissue (brown color); and (**b**) androgen receptor negative breast cancer. There is no brown color staining in androgen receptor negative breast cancer tissue.

**Figure 2 medsci-04-00015-f002:**
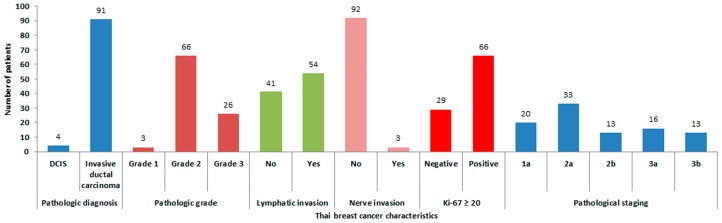
Thai breast cancer patients’ characteristics and staging in this study. DCIS: ductal carcinoma in situ.

**Figure 3 medsci-04-00015-f003:**
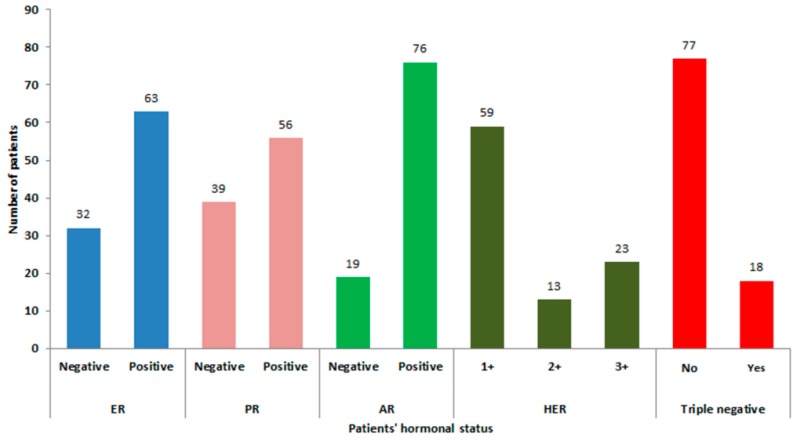
Hormonal status of Thai breast cancer patients in this study. ER: estrogen receptor; PR: progesterone receptor; AR: androgen receptor; HER: human epidermal growth factor receptor.

**Figure 4 medsci-04-00015-f004:**
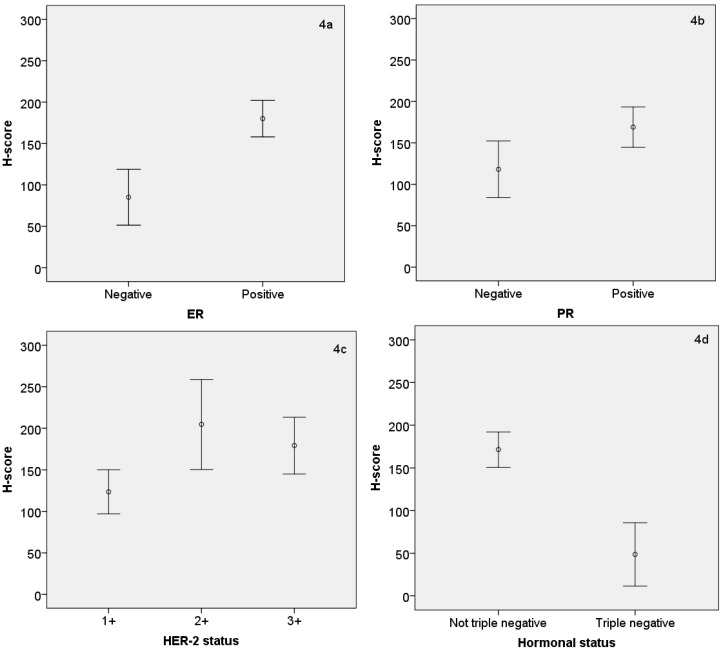
(**a**–**d**) Error bar graph showing the mean ± standard error (S.E.) of androgen H-score by ER, PR, HER-2 and hormonal status. The means ± S.E. were as follows: 85.2 ± 16.6 in ER^−^ and 180.1 ± 11.0 in ER^+^; *p* < 0.001; 118.1 ± 16.9 in PR^−^ and 169.0 ± 12.1 in PR^+^; *p* = 0.014; 123.6 ± 13.0 in HER-2 1+, 204.6 ± 24.8 in HER-2 2+ and 179.1 ± 16.5 in HER-2 3+; *p* = 0.006; 171.4 ± 10.3 in any hormone receptor positive group and 48.6 ± 17.6 in hormone receptor triple negative group; *p* < 0.001.

**Figure 5 medsci-04-00015-f005:**
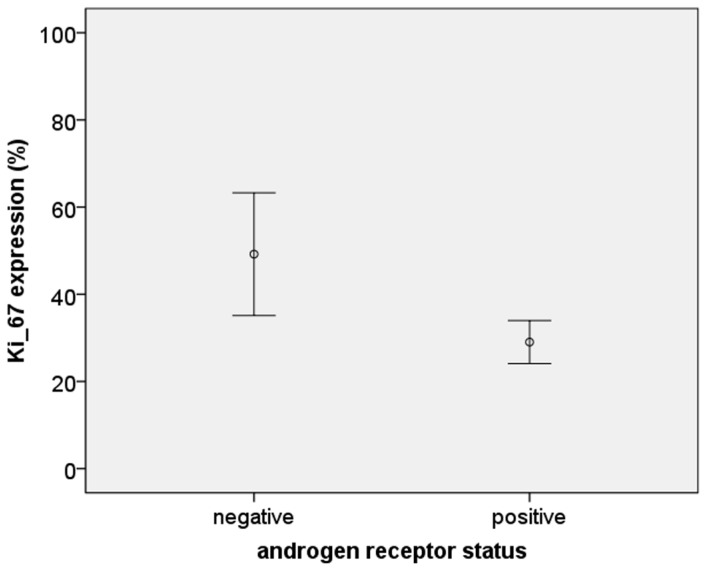
Error bar graph showing the mean ± S.E. of Ki-67 expression by androgen receptor status (49.21 ± 6.70 in AR negative group and 29.03 ± 2.48 in AR positive group; *p* = 0.014).

**Table 1 medsci-04-00015-t001:** Univariate analysis of factors related to androgen receptor (AR) expression.

Factors		Androgen Status		*p* Value
		Negative (*n* = 19)	Positive (*n* = 76)	
Menstrual status	Pre-menopause	13	30	0.092
	Peri-menopause	1	10	
	Menopause	5	36	
ER	Negative	15	17	<0.001
	Positive	4	59	
PR	Negative	14	25	<0.001
	Positive	5	51	
HER2	1+	17	42	0.022
	2+	1	12	
	3+	1	22	
Ki-67 ≥20	Negative	2	27	0.034
	Positive	17	49	
Hormonal status	Not triple negative	7	70	<0.001
	Triple negative	12	6	

ER: estrogen receptor; PR: progesterone receptor; HER: human epidermal growth factor receptor.

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
