# Peer review of "Androgen Receptor Expression in Thai Breast Cancer Patients"

_medsci, 2016, doi:10.3390/medsci4030015_

Reviewer 1 Report

General observation

The aim of this study was to find the prevalence of AR expression as well as the associated clinical and pathological features. You concluded that AR is inversely correlated with Ki-67, but you didn’t show the clinical value of such finding. Are these patients AR expressers presenting slower tumor progression, slower or lesser metastasis rate? How about the case-fatality in AR expressing patients? Since AR implication in breast cancer biology is somewhat ambiguous, it is relevant to deepen the study by providing more information about the clinical outcomes as well.

Introduction

Lane 36: when the author stated “the total number of breast cancer cases in Thailand …”, one would expect to read the prevalent but not the incident cases (new cases).

Lane 52: Here, “high mortality rate” may actually refer to “case-fatality” (death over total number of cases)

Ethical considerations

Lane 88: I am just curious about the obligations? Who are they for? The patients or the investigators? If they are for the patients, what are these obligations that you stipulated in the informed consent?

Data analysis

You did not perform descriptive statistics exclusively; bivariate and multivariate analysis are considered to be inferential statistics. Please correct this section.

References

There are too many references because you doubled or even tripled them.

Author Response

Manuscript ID: medsci-141897

Title: Androgen receptor expression in Thai breast cancer patients

Author response: We would like to thank the editor and reviewers for very positive and constructive comments, and providing us with the opportunity to address them. Furthermore, it should be noted that some, by itself, relevant questions involve studies that may require many years of follow up.  Please find our answers to your questions below.

Reviewer comments for authors:

1st Reviewer

Comments and Suggestions for Authors

General observation

The aim of this study was to find the prevalence of AR expression as well as the associated clinical and pathological features. You concluded that AR is inversely correlated with Ki-67, but you didn’t show the clinical value of such finding. Are these patients AR expressers presenting slower tumor progression, slower or lesser metastasis rate? How about the case-fatality in AR expressing patients? Since AR implication in breast cancer biology is somewhat ambiguous, it is relevant to deepen the study by providing more information about the clinical outcomes as well.

Author response: Thank you for a valuable suggestion. We do follow up on the patients, but obviously the mortality risk is also modified by the treatment. Two patients in the androgen positive group died in three years follow up period. This is not statistically significantly different when compared to the androgen negative group. We have added this statement to the discussion (page 8, line 252-254): “considering the limited time that has passed since the collection of the tissue samples in this study, no conclusions can be drawn on clinical outcomes and prognosis in our study population based on AR status.” The other variables, as stated in our manuscript, did not statistically relate to androgen status such as staging or tumor grading.

Introduction

Lane 36: when the author stated “the total number of breast cancer cases in Thailand …”, one would expect to read the prevalent but not the incident cases (new cases).

Author response: We apologize for any confusion. The numbers given indicate the incidence (new cases). The sentence has been modified. It now reads “The number of new breast cancer cases in Thailand in 2012 was about 12,000 cases, which accounts for 137.5 per 100,000 persons/year” (page 1, 36-37).

Lane 52: Here, “high mortality rate” may actually refer to “case-fatality” (death over total number of cases)

Author response: Indeed, “case fatality rate” is the correct term to use here. This has been changed in the text.

Ethical considerations

Lane 88: I am just curious about the obligations? Who are they for? The patients or the investigators? If they are for the patients, what are these obligations that you stipulated in the informed consent?

Author response: There were no obligations for the patients other than to acknowledge receipt of all information regarding the study. Neither was there any “responsibility” for the patient (same sentence). In order to avoid any confusion, we have removed these two statements. It now reads: “In addition, these patients received full information and explanation about the study objectives and rights that the patient has before enrolling in the study”.

Data analysis

You did not perform descriptive statistics exclusively; bivariate and multivariate analysis are considered to be inferential statistics. Please correct this section.

Author response: Thank you for this correction. Terminology has been changed in the text as “The statistical analysis of the clinical and pathological data was done with descriptive and inferential statistics.”.

References

There are too many references because you doubled or even tripled them.

Author response: One duplicate reference was removed. Several references have been removed in order to reduce the number of references. The total number is now 54.

Reviewer 2 Report

Major concerns:

1. Authors do not specify the ethnicity/racial identity of the patient population. Does "Thai"  breast cancer patients imply that all the patients were genetically/ racially Thai? Location of a study does not necessarily imply homogeneity of study population unless meticulously recorded and clearly stated.

2. It is unclear what the authors want to indicate through the following sentence: "immunohistochemistry for AR is not standardized."  If that is the case a study based on just that methodology is difficult to draw conclusions from. The authors should state what the other methods for detection of androgen receptors are and use at least one other detection method on at least one patient sample as representative to show similar readout between different detection method to establish confidence in their study result.

3. Authors should include another figure with bar graphs showing the difference in receptor expression profile between the Thai vs the Caucasian populations.

Minor concern:

Please remove the lines from the background of the graphs for cleaner representation of data.

Author Response

Manuscript ID: medsci-141897

Title: Androgen receptor expression in Thai breast cancer patients

Author response: We would like to thank the editor and reviewers for very positive and constructive comments, and providing us with the opportunity to address them. Furthermore, it should be noted that some, by itself, relevant questions involve studies that may require many years of follow up.  Please find our answers to your questions below.

2nd reviewer

Comments and Suggestions for Authors

Major concerns:

1. Authors do not specify the ethnicity/racial identity of the patient population. Does "Thai"  breast cancer patients imply that all the patients were genetically/ racially Thai? Location of a study does not necessarily imply homogeneity of study population unless meticulously recorded and clearly stated.

Author response: Thank you for this rightful comment. Indeed, we did not determine genetic descent and considered all women of Thai ethnicity as “Thai”. Clearly, that does not imply genetic homogeneity. We have added a statement on this in the discussion section (page 7, lines 208-211): “Here, genetic variations might also play a role. It should be noted that Thailand’s population is relatively heterogeneous culturally, but genetically consists of only a few ethnic groups of which some have Chinese, Vietnamese or Malaysian backgrounds. In this study, we did not establish genetical descent, but considered all women of Thai ethnicity as “Thai”.”

2. It is unclear what the authors want to indicate through the following sentence: "immunohistochemistry for AR is not standardized."  If that is the case a study based on just that methodology is difficult to draw conclusions from. The authors should state what the other methods for detection of androgen receptors are and use at least one other detection method on at least one patient sample as representative to show similar readout between different detection method to establish confidence in their study result.

Author response: We are sorry for the misunderstanding. What was meant was that while immunohistochemistry is a common method for protein detection, there is no standard cut off point for androgen expression in breast cancer. Therefore, we adopted the AR staining procedure that is used for prostate cancer and the H-score method, which is similar to the immunohistochemical scoring methods that are used for other hormone receptors. This statement is now added to the methods section (page 3, lines 100-104).

3. Authors should include another figure with bar graphs showing the difference in receptor expression profile between the Thai vs the Caucasian populations.

Author response: Although we agree that this would be really interesting, this was not a focus of our study. Also, we did not determine AR expression in Caucasian population(s) ourselves. However, in the discussion section, receptor expression profile differences are addressed thereby including ethnic differences. We feel that adding a figure with only our study results and Caucasian populations would be too limited and should also span other ethnicities, such as Hispanic or African-American. This comparison, however, would entail more than just adding numbers to a figure, but also would entail comparison of study (patient) population, experimental set-up and statistical analysis. Although from a scientific and clinical point of view very relevant, we hope that the reviewer can accept that this is clearly beyond the scope of our current study.

Minor concern:

Please remove the lines from the background of the graphs for cleaner representation of data.

Author response: The lines in graph background were removed.

Round  2

Reviewer 1 Report

The authors provided reasonable answers to my concerns. My opinion is favorable for a publication of the manuscript as such. 

Reviewer 2 Report

NONE